# Community-Engaged Development of Strengths-Based Nutrition Measures: The Indigenous Nourishment Scales

**DOI:** 10.3390/ijerph21111496

**Published:** 2024-11-11

**Authors:** Tara L. Maudrie, Laura E. Caulfield, Cassandra J. Nguyen, Melissa L. Walls, Emily E. Haroz, Laura R. Moore, Rachel G. Dionne-Thunder, Joe Vital, Brook LaFloe, Alanna Norris, Vincent Dionne, Virgil Pain On Hip, Jessica Dickerson, Kerry Hawk Lessard, Antony L. Stately, Valarie Blue Bird Jernigan, Victoria M. O’Keefe

**Affiliations:** 1Center for Indigenous Health, Department of International Health, Bloomberg School of Public Health, Johns Hopkins University, Baltimore, MD 21205, USA; 2Program in Human Nutrition, Department of International Health, Bloomberg School of Public Health, Johns Hopkins University, Baltimore, MD 21205, USA; 3Department of Nutrition, University of California Davis, Davis, CA 95616, USA; 4Alexandria City Public Schools, Alexandria, VA 22314, USA; 5Indigenous Protector Movement, Minneapolis, MN 55404, USA; 6East Phillips Neighborhood Institute, Minneapolis, MN 55404, USA; 7Niniijaanis One of Ones, St. Paul, MN 55106, USA; 8Native American Lifelines, Baltimore, MD 21202, USA; 9Native American Community Clinic, Minneapolis, MN 55404, USA; 10Center for Indigenous Health Research and Policy, Oklahoma State University Center for Health Sciences, Tulsa, OK 74135, USA

**Keywords:** nutrition, American Indian/Alaska Native, measure development, community based participatory research, mixed methods, indigenous, food sovereignty

## Abstract

Mainstream approaches to nutrition typically focus on diet consumption, overlooking multi-dimensional aspects of nutrition that are important to American Indian/Alaska Native (AI/AN) communities. To address health challenges faced by AI/AN communities, strengths-based measures of nutrition grounded in community worldviews are needed. In collaboration with AI/AN communities in Baltimore and Minneapolis, we developed the Indigenous Nourishment Scales through three phases. Phase 1 involved focus group discussions with nine community-research council (CRC) members (*n* = 2) and four in-depth interviews (*n* = 4) to gather perspectives on existing models of nutrition. Phase 2 refined scales through two additional focus group discussions (*n* = 2) with a total of nine participants and two in-depth interviews (*n* = 2). Finally, in Phase 3, we held ten (*n* = 10) cognitive interviews with AI/AN community members to refine the scales. Participants appreciated the measures’ ability to provoke reflection on their relationship with nutrition and suggested adjustments to better capture cultural nuances, such as incorporating concepts like “being a good relative” to land. The Indigenous Nourishment Scales represent a departure from conventional approaches by encompassing multiple dimensions of nourishment, offering a framework that addresses epistemic injustices in nutrition measurement and grounds health measurement efforts directly in community perspectives and worldviews.

## 1. Introduction

Knowledge systems, or epistemologies, that inform research are inherently embedded in cultural contexts and worldviews. Often, the knowledge systems that inform the development and interpretation of quantitative measures are rooted in Western, Eurocentric knowledge systems, and are primarily concerned with validity and reliability as they are defined by Western science [1,2]. However, communities have their own epistemologies and cultural contexts that are equally important for developing quantitative measures that accurately reflect their worldviews, values, and practices related to health and wellbeing [1,2].

American Indians and Alaska Natives (AI/AN) are extremely diverse populations, yet they share a common history of settler colonialism resulting in loss of lands, disruption of cultures, languages, and foods [3]. AI/AN communities hold unique values, knowledge, and worldviews shaped by centuries of living in reciprocity with their homelands. However, settler colonial societies intentionally sought to denigrate and erase these Indigenous ways of living, aiming to delegitimize Indigenous knowledge systems and sever the relationships between Indigenous peoples, their communities, and the land [4,5]. This historical marginalization persists today, as Indigenous knowledge and worldviews are often ignored, overlooked, and disregarded in research spaces, perpetuating ongoing epistemic injustices and violences against Indigenous peoples.

Key differences between Indigenous epistemologies and ontologies (or ways of knowing and being) and Western paradigms are rarely acknowledged, especially in the biomedical sciences (including nutrition). Western epistemologies, which dominate scientific inquiry, emphasize objective approaches to reduce perceived bias and have constructed rigid boundaries around what is considered “real” knowledge [6,7]. These boundaries have often excluded Indigenous knowledge systems, such as place-based, intergenerational, and intuitive forms of knowing, paralleling the settler-imposed borders on Turtle Island that served to exclude, dispossess, and dehumanize Indigenous peoples [6,8]. By reinforcing these boundaries, dominant research traditions have made Western epistemologies so pervasive that they are nearly universally accepted as the only valid knowledge system and way of approaching research [9]. In contrast, Indigenous epistemologies are inherently place- and relationship-based, guided by relational responsibility to place, to fellow humans, and to more-than-human relatives (plants, animals) [8,10,11]. Further, Indigenous epistemologies recognize that knowledge is relational, and knowledge is valuable not for its applicability to other contexts or to avoid the appearance of bias in a Western sense, but for its ability to forge and strengthen relational connection and understanding [10]. Indigenous ways of knowing and being challenge dominant research paradigms and assert the need for diverse knowledge systems that value relationality over objectivity. To truly address the epistemic injustices perpetuated against AI/AN communities, researchers must embrace Indigenous knowledge systems as legitimate and necessary, recognizing the profound insights they offer for both Indigenous and non-Indigenous contexts.

Despite enduring historical and ongoing traumas (including epistemic injustices and violences), AI/ANs in the United States continue to thrive and innovate, asserting the importance of their worldviews in contemporary spaces, including in research [12,13,14]. Through community- and tribal-based participatory research (CBPR) AI/AN communities have led the way in re-defining relevant health constructs (e.g., quality of life, wellness) and in developing measures that reflect their community worldviews and priorities [15,16,17,18,19]. As succinctly stated by Walls and colleagues, “measurement matters” especially in AI/AN communities where existing measures, even if they are deemed ‘rigorous’, may not function as intended or have been evaluated for cultural fit [12].

The field of nutrition is complex, and diet, often considered to be a primary nutrition exposure and/or outcome, is not a singular entity, but rather a dynamic composition of the foods and beverages consumed over time [20]. At the individual level, a person’s diet is strongly influenced by their approach to eating, which is shaped by a variety of factors such as food attitudes and beliefs (including mindfulness and intuitive eating), social norms, nutrition literacy, environment, and household food security status [21,22,23,24,25,26]. Many of these nutrition-related factors and nutrition beliefs are undoubtedly influenced by cultural context, worldviews, and values [27,28,29]. For instance, in the United States, foods are often categorized as ‘good’ or ‘bad’, and the choices made by food-insecure families are frequently judged as ‘poor’, rather than being understood as survival-driven decisions [30,31]. Indeed, food is deeply rooted in many cultures, with food and meals often serving as a vehicle to deliver cultural values, teachings, intergenerational knowledge, and family and community bonding [27,29]. Despite the role of culture in shaping approaches to eating and overall nutrition, typical measurement efforts focus on nutrient intakes or food and beverage patterns assessed through standardized metrics. This approach can overlook the nuanced cultural and contextual factors that influence approaches to eating and nutrition outcomes.

Understanding the cultural and contextual factors impacting approaches to eating is of particular importance for AI/AN communities whose nutrition, and subsequently their health, have been compromised by centuries of settler colonialism and structural racism [32,33]. Given the high prevalence of type 2 diabetes and cardiovascular diseases in AI/AN communities, diet is often a primary outcome of research interventions and a focus of public health programming [34,35]. Recently, some efforts have documented the importance of culture within nutrition in AI/AN communities by measuring food sovereignty, engagement with AI/AN food practices, and traditional food consumption [36,37,38,39,40,41]. However, many of these measures focus heavily on dietary intake, with less attention paid to cultural or psychosocial factors. Though there is value in conventional assessments of dietary intake for identifying and addressing health inequities, there is also a need for rigorous strengths-based measures that capture a variety of nutrition-related factors providing a complete picture of AI/AN nutrition [42,43,44].

Nourishment is defined by the *Oxford English Dictionary* as “something which nourishes or sustains; sustenance, food” [45]. The concept of nourishment is more expansive than simple nutrition, which refers to only the act of consuming and utilizing food substances for physiological purposes. For AI/ANs, the concept of nutrition captures holistic aspects of wellbeing that extend beyond food sources. For AI/ANs, the concept of nutrition can include aspects of physical, emotional, relational (social), and spiritual health, making it more aligned with the concept of nourishment than with the narrower definition of nutrition typically used in the biologic sciences and in the field of dietetics [29,46,47,48].

In response to calls from Indigenous communities for strengths-based approaches to nutrition, the Indigenous Nourishment Model (INM) was developed through qualitative research with AI/AN and Native Hawaiian food experts. It aims to assess important aspects of nourishment for AI/ANs that may not be captured through existing conceptual models and measures [46]. The INM provides a framework for understanding and measuring nutritional influences through a holistic strengths-based lens rooted in cultural teachings and values. The INM consists of four main domains of nourishment: physical, spiritual, emotional, and relational, as well as intersecting components of nourishment that illustrate the complexity of Indigenous nourishment practices and beliefs (Figure 1; [46]). Through visual representation of intersections between domains of nourishment, the INM emphasizes the holistic nature of nourishment and the importance of considering multiple dimensions of wellbeing in nutrition research and practice. This model serves as a promising framework from which to develop AI/AN strengths-based nutrition measures aligning with Indigenous worldviews.

In this study, we describe an iterative community-engaged approach to developing a set of measures based on the INM. Our overall aim is to improve the measurement of holistic nutritional health to ground the measurement and evaluation of nutrition- and health-related interventions in AI/AN worldviews.

## 2. Materials and Methods

The current study focuses on measure development as part of a larger sequential exploratory mixed-methods Community-Based Participatory Research (CBPR) investigation of nutrition in AI/AN communities and the lead author’s (Tara L. Maudrie; T.L.M.), dissertation research [46]. This study took place in two urban locations: Minneapolis and Baltimore. Minneapolis is home to more than 35,000 urban AI/ANs and more than four AI/AN health organizations, including Native American Community Clinic (NACC). NACC is a federally qualified health care center that provides medical, dental, behavioral, and spiritual health care. The Baltimore metropolitan area is home to more than 14,000 AI/ANs and is served by one health organization, Native American Lifelines (NAL), which is Title V Indian Health Service contracted Urban Indian Health program that serves both the Baltimore and Boston metropolitan areas. The first author, Tara Maudrie has collaborated with NACC and NAL through research and volunteer work for more than four years. The research described in this manuscript reflects ongoing partnerships with both communities and community partner organizations. The overall study design was co-determined with the executive leadership and staff of community research partner organizations prior to applying for research funding.

We used a CBPR approach to guide our measure development approach and study implementation. Community research councils (CRCs) were created for this study in collaboration with our community research partners, NACC and NAL. Each community had their own CRC which served other purposes for partner organizations. The composition of the CRCs was determined in collaboration with partner organizations, and members were AI/AN adults purposively identified by partner organizations [49]. Purposive selection of CRC members by community research partners was utilized to gather a group of well-rounded community members who represented various positionalities and areas of expertise in order to effectively inform and guide ongoing research, including the current study. Purposive sampling was chosen over a simple random sample to ensure that the CRC would be composed of individuals who could contribute rich, context-specific perspectives, strengthening the relevance and cultural grounding of the project. Membership eligibility included being a member of their respective community, being over 18 years of age, and being a self-identified AI/AN. Throughout the duration of the research study, the number of CRC members fluctuated to accommodate members’ time conflicts and busy workloads, and to add new members to round out the knowledge and experiences of existing CRC members.

Our CBPR measurement development approach involved three major phases (Figure 2):Phase 1: Gather CRC perspectives to improve the Indigenous Nourishment Model [46], collect feedback on other approaches to nutrition, and to provide direction for measure development, including the drafting of survey items.Phase 2: Gather CRC feedback on item wording, content, and response options of draft measures developed from the previous phase.Phase 3: Pilot test newly refined measures through cognitive interviews with purposively selected urban AI/AN adult community members.

In the current study, CRC members made up our study sample for Phases 1 and 2 of the research. While some CRC members participated in both Phases 1 and 2, some participated in only one phase due to scheduling conflicts and busy workloads outside of their CRC commitments. As described later in the Methods section, CRC members also contributed to recruitment of AI/AN adults for cognitive interviews in Phase 3. All focus group discussions (FGDs), in-depth interviews, and cognitive interviews were conducted by the lead author (T.L.M.), who is an Indigenous researcher trained in qualitative and mixed research methods. In-depth interviews and FGDs were audio-recorded and transcribed verbatim. Analytic memos were composed following each data collection event and throughout iterative analysis of data. Analytic memos identified important themes and considerations for measure development and served as a way for the lead author to critically reflect on her own role in the research process. Prior to the in-depth interviews, FGDs, and cognitive interviews, all participants were provided consent information via email or physical mail, and all participants verbally consented to participation at the time of data collection. Ethical approval was received from the Johns Hopkins Bloomberg School of Public Health Institutional Review Board (IRB00026314) and community partner organizations through their involvement and sign-off on project plans.

### 2.1. Phase 1: Feedback on Models of Nutrition, and Guidance for Measure Development Through FGDs and In-Depth Interviews with CRC Members

CRC members from both CRCs were invited to participate in Phase 1 of the research via email and written physical invitations when internet access was a limiting factor. Following consent, CRC members listened to a thorough review of the INM and the research that contributed to developing the INM, including relevant qualitative quotes from the prior research (see Figure 1) [46]. CRC members provided general feedback on the model and iterative probing was used to gather nuanced input on the utility, resonance, and application of the model to urban AI/AN communities. Following review of the INM, CRC members also reviewed and provided feedback on three mainstream approaches to eating: intuitive eating [50], holistic nutrition [51], and mindfulness eating [52]. The purpose of reviewing additional mainstream models of eating was to understand if aspects of these models resonated with urban AI/AN communities to inform revisions of the INM and subsequent measure development. intuitive eating, holistic nutrition, and mindfulness eating were chosen as models to review with CRCs as they each approach nutrition holistically, incorporating aspects of mental and social wellbeing, which may have potential resonance with AI/AN worldviews.

Two virtual (*n* = 2) FGDs were conducted with two CRC members in the first FGD and three CRC members in the second FGD. FGD 1 was 85 min in length (1 h and 25 min), while FGD 2 was 77 min (1 h and 17 min). Four CRC members completed in-depth interviews s (*n* = 4) due to scheduling conflicts with FGDs. One in-depth interview was conducted in person, while the remaining three were conducted virtually. The in-depth interviews ranged in length from 49 to 68 min (average: 56 min). Transcripts were open-coded by author T.L.M. to identify relevant feedback for model refinement and measure development. Each CRC member received a gift card valued at $100 USD as a token of appreciation for their participation in this phase.

### 2.2. Phase 2: Measure Development and Feedback on Developed Measures with CRC Members

We developed a list of important concepts to measure and pieces of the conceptual model to refine based on Phase 1 findings. An initial list of 43 draft items was generated corresponding to dimensions of the INM and feedback from CRCs. Through an iterative process guided by co-authors (Authors L.E.C., M.L.W., E.E.H., & V.M.O.) and scale development guidelines, author T.L.M. refined the initial list of draft items, organizing the set of items into two distinct scales based on item content [53]. The first draft scale assessed access to, and participation in, Indigenous food practices. The second scale assessed psychosocial aspects of nourishment, including beliefs and values.

As in Phase 1 of the research, CRC members from both CRCs were invited to participate in Phase 2 of the research via email and written physical invitations when internet access was a limiting factor. During FGDs and in-depth interviews, CRC members pilot tested the two draft scales. Using open-ended questioning and iterative probing, CRC members were asked to describe their experience completing the draft survey items and elaborate on any parts they found difficult, hard to understand, or unnecessary and if they felt as if any important and relevant topics were not covered in the survey. CRC members were also asked to discuss how they felt the items would be received by members of their community (e.g., item comprehension and cultural resonance).

During this phase of data collection, one virtual FGD (*n* = 1) was conducted with 3 participants, one in-person FGD (*n* = 1) was held with 6 participants, and two (*n* = 2) virtual in-depth interviews were conducted. FGD 1 was 59 min in length, FGD 2 was 85 min (1 h and 25 min), and in-depth interview 1 was 56 min in length, while in-depth interview 2 was 50 min in length. Again, each participating CRC member received a gift card valued at $100 USD for their participation. Transcripts from Phase 2 in-depth interviews and FGDs were open-coded and important suggested changes and feedback were identified as themes. To organize feedback from CRC members, throughout data collection and analysis author T.L.M. composed memos, and tracked suggested changes and feedback in a joint display. A joint display is a mixed-methods strategy that merges qualitative and quantitative findings and achieve meta-inferences [54]. This joint display table was organized by rows with the wording of Phase 2 draft measure instructions, items, and response options, and columns for CRC feedback, suggested changes, and eventually a column for revised item wording. During this phase, the two original scales were split into three distinct scales with a total of 48 items and used for subsequent cognitive interviews. Scale A, “Relationships with Food”, included 13 items paired with a 7-point Likert scale (*strongly disagree* to *strongly agree*) intended to assess one’s holistic relationship with food. Scale B, “Indigenous Food Values and Beliefs”, contained 18 items designed to assess Indigenous food values and available response options also used a 7-point Likert scale (*strongly disagree* to *strongly agree*). Finally Scale C, “Indigenous Foodways Engagement and Access Satisfaction Inventory”, contained 17 items which measured frequency of participation in and access to Indigenous food practices with the following response options: never, once or twice, seasonally, monthly, weekly, or daily. For Scale C, items refer to access and participation in the past year; this timeframe was chosen due to the seasonal nature of Indigenous foodways and because some Indigenous food practices are only practiced during a certain period of the year.

### 2.3. Phase 3: Cognitive Interviews with Urban AI/AN Adult Community Members

Cognitive interviews were conducted to gain a deeper understanding of how items and response options were interpreted at the individual level and to guide further revisions of scales from members of the target population. Cognitive interviews are a method used to identify and correct problems with measures and scales [55,56]. Adult AI/ANs in both communities were identified as potential participants through communication with CRCs and executive leadership of partner organizations and were contacted through email and provided more information about participation in virtual cognitive interviews. As the interviews progressed, the lead author communicated with CRCs and executive leadership to identify and recruit individuals representing a range of positionalities and lived experiences (e.g., age, gender, tribal affiliations, literacy levels) to ensure a well-rounded and diverse sample. Prior to interviews, each participant was given a verbal consent script and provided an opportunity to ask questions before giving verbal consent. Participants were reminded that the purpose of the interview was to learn from their experience taking the draft survey and to gather feedback on potentially confusing aspects of the scales [57]. Participants were encouraged to use the “think out-loud” approach as they took the survey, meaning that participants were asked to verbalize their thoughts as they read and responded to items [57]. Throughout the interview, probes included interpretation probes like, “*Can you tell me more about how you came up with your answer?*” and paraphrasing probes like, “*What does [term] mean to you in the context of this question?*” to engage participants in the think out-loud process and to elicit specific feedback about item content and language, and identify potential problems from interviewees [57].

Ten (*n* = 10) virtual cognitive interviews were completed, with adult (18 years or older) AI/ANs who were members of their respective communities (Minneapolis, MN, USA, St. Paul, MN, USA, or Baltimore, MD, USA) and represented a range of positionalities through varied tribal affiliations, age and gender diversity, and community roles/involvement. Interviews varied in length, with the shortest being 25 min and the longest being 85 min; on average, interviews were 46 min in length. All cognitive interviews were audio-recorded and transcribed verbatim. Transcripts were open-coded to find and organize relevant pieces of feedback. Each participant received a gift card valued at $50 USD as an incentive for providing their time and feedback.

## 3. Results

### 3.1. Final Indigenous Nourishment Scales

As described in the methods, our approach to measure development followed an iterative community-engaged study design, which resulted in three distinct measures of Indigenous Nourishment. The final content of these scales and their response options can be found in Table 1. The remainder of the results section describes the rich feedback and revisions from each phase of the research which contributed to these final scales (Table 1) and the graphically designed Indigenous Nourishment Model presented earlier in the introduction (Figure 1).

### 3.2. Phase 1: Feedback on Models of Nutrition and Guidance for Measure Development Through FGDs and In-Depth Interviews with CRC Members

All CRC members expressed appreciation and support for the Indigenous Nourishment Model, remarking on aspects of the model that reflected their own personal experiences and the experiences of their communities. In an FGD, a male CRC member shared, “I go back to the four quadrants that were highlighted at first, food sharing, ancestral memory, belonging, and intentionality. **Those are just core values** I think Indigenous people carry in their every day to day. That’s why I think folks **will be able to see themselves in this model**”.

Several members offered feedback for improving the model, including working with an artist or graphic designer to make the model more esthetically pleasing and clarifying that food sharing includes sharing knowledge about food. For example, in an in-depth interview, a male CRC member shared, “I think visual diagrams are really important too, just for people to really visualize what it is that you’re conveying. And I think that this does a good job of doing that. But I’ve seen things similar to this with the four directions too... I could totally see something like this a **little bit more polished on a poster or on a flyer**”. A female CRC member communicated that she felt like the educational part was missing. She voiced this by stating, “One thing I feel like might be missing, and it’s kind of touched in original instructions and connection across time, is **how do we educate people**? How do we educate our own communities? Some people don’t even know where their ancestral lands are, so how will they know what [ancestral] foods were available to them”? To address these points of feedback, the wording in the model was adjusted to reflect that food sharing encompasses physical food sharing and food knowledge sharing, which can be educational. The lead author also worked with an Indigenous artist, Kenrick Escalanti, to graphically design a version of the model (Figure 2).

Although CRC members expressed appreciation for aspects of other nutrition models, like the flexibility provided through intuitive eating [58], limitations of these models were identified. These included the lack of applicability in low-income settings and grounding in Western concepts. In an FGD, a male CRC member shared that he did not believe intuitive eating had much utility in his community. He elaborated saying, “When I see this, I don’t get how poor folks [in our community] are supposed to practice intuitive eating **when access is so limited**. This is for yups or folks who have the means to be like, ‘Oh yeah, I can eat this honeybun because it makes me feel good, but eventually I’m going to eat a salad for sure’. I understand the thinking behind this because it challenges the idea that you don’t have to eat healthy every day, just listen to your body… In the Venn diagram [Indigenous Nourishment Model] before, all [social] class participants can participate because there’s **the social and the spiritual connection to the nourishment**, where this [Intuitive Eating]… **I know what intuitive eating is, but it’s not something that I think my family could understand because of the class differences**”. In regard to the holistic nutrition model, another male CRC member succinctly stated, “This is just… Yeah, it’s a **Western concept of holisticness**”.

All CRC members who participated in Phase 1 agreed that the Indigenous Nourishment model would be appealing to people in their community and that it was easy to understand. In an FGD, a female CRC member expressed her appreciation for the more expansive approach to nutrition. She said, “The nutrient focus is also important, but it’s also super important not to block out that community healing aspect of food and **how it brings us together**… That [model] doesn’t have to be updated every year like the dietary guidelines, that’s always going to be a constant thing that’s in our minds, that comes from the heart too, which is really nice. And I feel like having a model, a concrete model like this out there to tie everything together, not only for us as Native people, but **even to inform other people who are non-Native of how we view food and how food is healing to us** could be just super big. So, I really hope that this does get out there and is incorporated into some of those larger models of nutrition that is shared nationwide or internationally or whatever it may be”. Another male CRC member responded to her point about nourishment being more than just nutrients by saying, “When talking about nourishment, I think in Western science it’s typically **just what you’re putting into your body**. And so, it puts into question the sustainability of that. Yeah, obviously we should all be eating leafy greens and berries and such, but those things are expensive as hell. But where the sustainability of this model comes is that the **connection with others.** You could be sharing, just growing up, it was spaghetti. Very simple, but if you’re sharing that with community, you’re getting that more holistic benefit to all of this. **There is a belonging because you’re having food with people**… but [this model] it’s a lot more sustainable because there’s much more brevity and cultural connectedness to this”.

Several CRC members remarked that, as we moved forward with the measure development process, considering food access was important. A female CRC member shared this concern by saying, “The first thing that came to my mind was asking about, **do you have access to your ancestral foods**? Do you even have access to promote these healing habits? Because I feel like you could get all of the answers by asking, ‘Do you feel connected to the foods that you eat’? **But if at the end of the day those people don’t have access to them, it’s not really a solution** in a sense, or it doesn’t really help them out”.

### 3.3. Phase 2: Measure Development and Feedback on Developed Measures with CRC Members

Overall, the initial items and scales were well received by CRC members. Several CRC members remarked that going through the items prompted them to think about food in a different way than they had previously and that they believed the items might have some cross-culture utility. In an FGD a male CRC member shared, “I like the questions. **I usually don’t get asked those type of questions** or I’m **not in those kinds of spaces to have those kinds of conversations.** But I feel like there’s a way you could structure the questions for specifically Natives and then non-Natives… **I haven’t thought about food like that in quite some time**. So, it was really cool to sit here and think about because lately I’m just too busy trying to survive and work, stay busy, and sometimes we forget about those teachings or how important food can be”. Responding to him, an Elder male CRC member shared, “I liked this, this is a document that can be addressed and in fact can be used as in aspects of other cultures, **bringing out an openness and an understanding of each other.** That is very good”. Although the scales were intended to have resonance within Indigenous communities, CRC members identified that the developed scales may have utility beyond AI/AN communities and may even promote cross-cultural understanding of food values.

Other CRC members expressed how they connected with particular items by providing context from their own lives. When reviewing an item related to belonging within a food system, a male CRC member shared, “I feel like when there’s a gathering, **I do feel like it’s a sense of belonging to share a meal with someone**, we’re in a space of vulnerability. Other than that, I usually don’t share my food. I won’t break bread unless I like you, unless I love you. But when we’re at a gathering or we’re in a ceremony, I definitely feel that connection and the spirit of it”. An Elder male CRC member responded to them, saying, “I strongly agree with that because helping people, **feeding people helps you connect with them** to where they can trust you”. Later, in the same focus group, a female CRC member communicated her appreciation for an item related to food sharing by stating, “I look at it [food sharing] as when we have our community suppers [in our local community], that’s the way that we can share food with others outside the household. And that’s why I appreciate them dearly when we have those… It lets us know that yes, **this is what we used to do, and we can still continue to do it**”. The positive responses and connections to various items expressed by CRC members indicated that items had face validity and cultural resonance with our two partner communities.

CRC members also offered specific suggestions for rewording items to be more aligned with Indigenous worldviews. For example, when reviewing an item related to stewardship of land, water, plants, and animals, three female CRC members shared the following discussion. A female CRC member shared, “I think my first thought is that you should add, ‘land or environmental stewardship’ in front of ‘stewardship’ on here. Just because **there’s other ways that you can steward things.** Like if you’re stewarding your donors, you know what I mean… But it’s a different thing if we’re talking about land and environment than social aspects of stewarding things”. Responding to her another female CRC Member said, “Yeah, maybe ‘steward’ is not the word. It’s like a **colonial term**, like I’m going to control and steward the land, the environment... How about, “**I have access to resources to be in balance with the environment**... Because it’s about balancing. It’s about balancing your impact versus what you were able to help maintain. Because that’s what our ancestors did for thousands of years. They weren’t out here like, ‘Oh, we’re going to steward the land’. Or ‘We’re going to manage, we’re going to control’. They were like, ‘How do we live in balance’?”. Finally, a female CRC member summarized their input and offered a specific wording change by saying, “I’ve heard people say, ‘**Good relative**’. Like be a good relative to the land, to the plants”.

### 3.4. Phase 3: Cognitive Interviews with Urban AI/AN Adult Community Members

Overall, the item content of scales was well received by participants. Similarly to participants in previous phases, participants in cognitive interviews often remarked that the items made them think differently. A male participant shared, “It was good hearing those questions. I have those questions, and **it’s weird that we don’t ask these questions**, so I guess **it’s comforting hearing these questions** and have a thought that somebody cares”.

Interviewer:“You mean somebody cares about what”?

Male Participant:“**Cares about the food**, the tradition of the food, the healthiness of the food, the nutrition, **the nourishment**”.

Respondents rarely had difficulties understanding the format of the scales or the instructions for each subscale. However, in the first two cognitive interviews, it became apparent that the strongly disagree to strongly agree Likert scale was not the best fit for the content of the items in Scale A. One participant explained this by saying “I feel like what I was looking for in terms of the answer scale would be a not an agree, disagree, but **rather like an occasion**”. Therefore, the remaining cognitive interviews were completed with the response options of never, rarely, occasionally, sometimes, often, very often, and always for Scale A. Additionally participants identified that one item in Scale A (“Through my traditional/ancestral foods I feel physically connected to my homelands”) fit better in Scale B. One Scale A item, “Food is a source of shame for me”, was particularly confusing for participants. Some participants connected this to the concept of eating as a way to cope with negative emotions, whereas others associated it with eating generally as being shameful, especially in households where families were concerned about weight or diabetes. As a result, this item was revised (“I feel proud of the foods I eat”) to match the strengths-based language used in the remainder of the items. In Scale B, several items were dropped, as they also had varied interpretations across participants. For example, an item worded “I can create a relationship with food, no matter where I am physically located”, prompted some participants to explain that they wouldn’t know how to survive on subsistence anywhere other than their homelands, while another participant noted, “it doesn’t matter where I’m at, I’ll always basically know what I like to eat”. Since the concept of connecting to food through land was covered by several other items, this item was dropped. Participants also identified items that would benefit from wording clarification. When reviewing an item worded as “Sharing meals with others in my community makes me feel connected”, a participant questioned, “So sharing meals with others in my community makes me feel **Connected to what**? Connected to community? Connected to food”? This item was intended to assess relational connection to community through food, and therefore was revised to read “Sharing meals with others makes me feel connected to my community”. In addition to these examples identified through cognitive interviews, other wording challenges, redundancies, and needed changes to items were identified and addressed. A sample of these revisions can be found in Table 2.

## 4. Discussion

For many years, the study of diet has predominantly focused on individual food consumption, and the alignment of these consumed foods with national dietary standards. However, food is essential in nurturing holistic wellbeing, including physical, spiritual, emotional, and relational health, which is not captured by typical dietary assessment tools [27]. This manuscript details an iterative community-engaged process to develop a set of nutrition-focused scales that resonate with AI/AN worldviews and offers new directions for approaching eating, and therefore nutrition, more holistically.

Our measure development process involved deep and iterative engagement of community members and members of community research councils in two urban Native communities. In the first phase, CRC members expressed how strongly they identified with the core values and themes represented in the Indigenous Nourishment Model and endorsed its potential for utility and resonance within their communities. Comparatively, existing approaches to food and eating like intuitive eating [58], mindfulness eating [52], and holistic nutrition [51], did not resonate with CRC members due to strong concerns about the lack of applicability within their communities, who experience problems with food access. The tension between psychosocial approaches to food and eating, especially intuitive eating, and food insecurity, including its limited applicability in socioeconomically disadvantaged settings, has been documented in the peer-reviewed literature [59,60]. Future research and public health practice efforts should continue to explore how AI/AN worldviews and definitions of nutrition could be integrated into dietary counseling. In subsequent phases, CRC and community members reviewed scales and provided important feedback to refine item and response wording. Our iterative community-engaged approach to measure development yielded three distinct scales capturing important indicators of Indigenous nutrition, including holistic relationships with food, cultural food values and beliefs, and engagement with and satisfaction with access to Indigenous food practices.

Our efforts align with initiatives to broaden the scope of nutrition-related measures, including developing measures of perceived nutrition environments (including community food environment) [22], indicators of food sovereignty [36,37], and creating culturally specific social determinants of nutritional health [34]. Moreover, our measures respond to calls from Indigenous communities for health measures that uplift community and cultural strengths [14,42]. The scales described in this manuscript were developed to elevate Indigenous epistemologies and strengths in part by focusing on cultural values including connections to food and the types of connectedness food facilitates, or as other scholars have termed it, ‘connectedness’ [61]. In recent years, Indigenous communities and researchers have made great strides in measuring consumption of traditional/ancestral foods, engagement with specific cultural food practices, and measuring cultural connectedness in conjunction with traditional food intake [40,41,62,63]. Our scales build upon and expand these efforts by aiming to comprehensively capture how cultural connectedness to food relates to holistic wellbeing.

Despite the novel contributions of our approach to measure development, this research is not without limitations. Our scales were developed in close collaboration with urban AI/AN adults, and although our participants represented a range of positionalities and tribal affiliations, it is unclear if these scales will resonate with AI/AN communities beyond our two partner communities. However, as CRC members noted, the Indigenous Nourishment Model and cales may indeed have resonance or transferability to other communities, including Indigenous populations nationally and internationally. Additionally, these measures were intended for adults and developed with AI/AN adults and Elders and have not been adapted for AI/AN youth. To be inclusive of age and developmental-specific needs, future research conducted in partnership with AI/AN youth is needed.

## 5. Conclusions

The Indigenous Nourishment Scales provide novel considerations on the meaning of nourishment for nutrition professionals and educators working to improve nutritional health in Indigenous communities. Along with proper nutrition education and food guidance literacy, the Indigenous Nourishment Scales may help them to adopt a more expansive view of nutrition than that offered through traditional measures of nutrition and to critical reflect on what it means to be nourished. Research is underway to evaluate the validity and reliability of the Indigenous Nourishment Scales with a sample of urban AI/ANs. The goal of these measure development efforts is to ground nutritional health measurement efforts directly in community and cultural perspectives.

## Figures and Tables

**Figure 1 ijerph-21-01496-f001:**
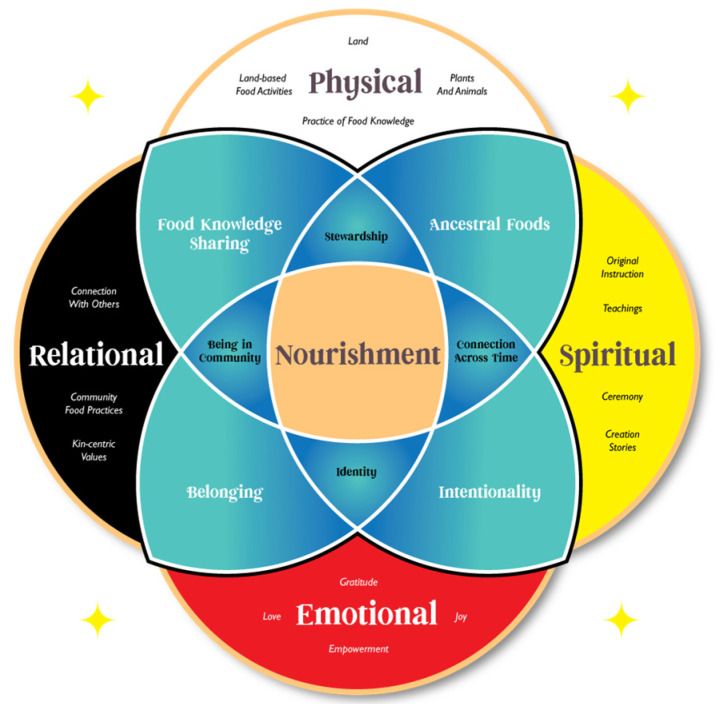
Graphically designed Indigenous Nourishment Model.

**Figure 2 ijerph-21-01496-f002:**
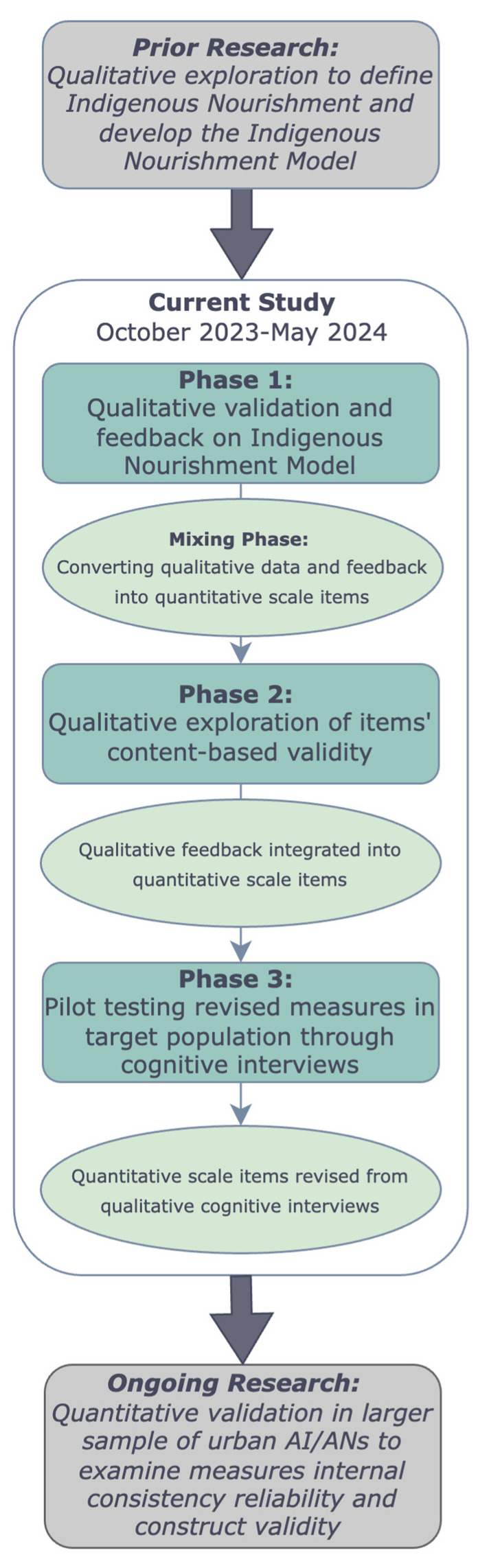
Sequential exploratory mixed-methods study design overview.

**Table 1 ijerph-21-01496-t001:** Final measures and item wording.

**Measure A: Relationships with Food** **^1^** This next set of questions will ask you about your relationship with food over the last year. Please select the frequency that best reflects your experience over the last year.	
1	Sharing food is a way I strengthen or build my relationship with others.
2	I feel emotionally nourished by the foods I eat.
3	Knowing where my food comes from makes me feel connected to that place.
4	I feel thankful for the foods I eat.
5	I feel proud of the foods I eat.
6	I feel physically nourished by the foods I eat.
7	I feel connected to plants, animals, waters and others through the foods I eat.
8	The foods I eat spiritually nourish me.
9	I feel physically connected to my homelands through my traditional/ancestral foods.
**Measure B: Indigenous Food Values and Beliefs** **^2^** The next section of this survey will ask you about your Indigenous food values and beliefs. Please rate each statement by how much you agree. Notes: Each Indigenous person has their own unique relationship to culture and food. For some people, these relationships can be complicated by various things like historical trauma, family histories, lack of food access, and even living in an urban community. Although this section of the survey asks about various traditional/ancestral food practices and values, we want to emphasize that your feelings and experiences as an Indigenous person are valid and important, regardless of whether you have or have not participated in traditional/ancestral food practices. This survey uses the term traditional/ancestral to refer to foods that have existed in Indigenous communities prior to colonization. It’s important to note that although cultural foods (e.g., fry bread, Indian tacos) are important to Indigenous communities, because they are made from ingredients not Indigenous to North America (e.g., wheat, dairy, beef, chicken) they are separate from traditional/ancestral foods (e.g., bison, venison, wild rice, ramps).	
1	Actively participating in my food system (e.g., tending to foods, feeding others, etc.) gives me a sense of belonging.
2	My traditional/ancestral foods are an important part of my cultural identity.
3	My traditional/ancestral foods connect me to future generations.
4	Sharing meals with others makes me feel connected to my community.
5	It is my responsibility to care for my food system, including plants, animals, and the world around me.
6	My spiritual beliefs influence my connection to food.
7	Eating foods that my ancestors ate makes me feel connected to them.
8	Participating in traditional/ancestral food practices (e.g., maple tapping, corn processing, meat processing, feasts) deepens my relationship with my environment.
9	I feel a responsibility to pass on traditional/ancestral food knowledge to future generations.
10	Traditional/cultural food practices (e.g., maple tapping, corn processing, meat processing, feasts) bring my urban Native community together.
11	Learning about or teaching others about Indigenous foodways brings me positive emotions.
12	I think of plants and animals as my relatives.
13	I am grateful for traditional/ancestral food knowledge that has been passed down intergenerationally in my community.
**Measure C: Indigenous Foodways Access and Participation Inventory** **^3^** This section of the survey will ask you how satisfied you are with your access to Indigenous food practices, and your participation in Indigenous food practices. Notes: Each Indigenous person has their own unique relationship to culture and food. For some people, these relationships can be complicated by various things like historical trauma, family histories, lack of food access, and even living in an urban community. Although this section of the survey asks about various traditional/ancestral food practices, we want to emphasize that your feelings and experiences as an Indigenous person are valid and important, regardless of whether you have or have not participated in traditional/ancestral food practices. This survey uses the term traditional/ancestral to refer to foods that have existed in Indigenous communities prior to colonization. It’s important to note that although cultural foods (e.g., fry bread, Indian tacos) are important to Indigenous communities, because they are made from ingredients not Indigenous to North America (e.g., wheat, dairy, beef, chicken) they are separate from traditional/ancestral foods (e.g., bison, venison, wild rice, ramps).	
1	In the last year, how satisfied were you with your access to Indigenous foodways (e.g., hunting, fishing, gathering, feasts, traditional forms of food processing, gardening, etc.)?
2	In the last year, how often did you participate in Indigenous foodways (e.g., hunting, fishing, gathering, feasts, traditional forms of food processing, gardening, etc.)?
3	In the last year, how satisfied were you with your access to traditional/ancestral foods?
4	In the last year, how often did you eat traditional/ancestral foods?
5	In the last year, how satisfied were you with your ability to be a good relative to your food system (e.g., tobacco to put down, composting, recycling, resources to tend to plants and animals, etc.)?
6	In the last year, how often did you intentionally take action to be a good relative to your food system (e.g., composting, putting down tobacco, tending to land, water, plants or animals, etc.)?
7	In the last year, how satisfied were you with your access to foods that are emotionally meaningful to you?
8	In the last year, how often did you eat foods that were emotionally meaningful to you?
9	In the last year, how satisfied were you with your access to Indigenous foodways (e.g., gathering, feasts, traditional forms of food processing, gardening, etc.) within your urban Native community?
10	In the last year, how often did you participate in Indigenous foodways (e.g., gathering, feasts, traditional forms of food processing, gardening, etc.) within your urban Native community?
11	In the last year, how satisfied were you with your ability to share food with others outside your household?
12	In the last year, how often did you share foods with others outside your household?
13	In the last year, how often did others outside your household share foods with you?
14	In the last year, how satisfied were you with your opportunities to share food knowledge (e.g., hunting, fishing, gathering, gardening, cooking, processing) with others in your urban Native community?
15	In the last year, how often did you gain new food knowledge (e.g., hunting, fishing, gathering, gardening, cooking, processing) from others in your urban Native community?
16	In the last year, how often did you share food knowledge (e.g., hunting, fishing, gathering, gardening, cooking, processing) with others in your urban Native community?

^1^ Response options: never, rarely, occasionally, sometimes, often, very often, always. ^2^ Response options: strongly disagree, disagree, somewhat disagree, neither agree nor disagree, somewhat agree, agree, strongly agree. ^3^ Items 1, 3, 5, 7, 9, 11, 14 response options: very dissatisfied, dissatisfied, somewhat dissatisfied, neither satisfied nor dissatisfied, somewhat satisfied, satisfied, very satisfied. Items 2, 4, 6, 8, 10, 12, 13, 15, 16 response options: never, once or twice, seasonally, monthly, weekly, daily (each of these items also included the option “I have never participated in XX” etc.).

**Table 2 ijerph-21-01496-t002:** Example revisions from Phase 3 cognitive interviews.

Cognitive Interview Item Wording	Cognitive Interview Feedback	Integration of Suggestions	Revised Item Wording
Actively participating in my food system (tending to foods, feeding others) makes me feel as if I belong to something larger than myself.	“I don’t see [how] if I’m not tending to foods, feeding others how I see myself as smaller as than before”.—CI 3	Reworded to infer a sense of belonging through food.	Actively participating in my food system (e.g., tending to foods, feeding others, etc.) gives me a sense of belonging.
It is my duty to care for my food system, including other people, plants, animals, and the world around me.	“It is my duty. I wonder if you could say my responsibility… I feel like duty feels more like job or work-based versus a responsibility is just as a mother or even personal outside of anything that is work-based too”.—CI 9	Reworded to remove the word duty.	It is my responsibility to care for my food system, including plants, animals, and the world around me.
Through food I feel connected to plants, animals, land, water, and others.	“This one also feels similar to [item 2.5B]…I actually prefer this one. I think because the language is a little bit more sparse and I think allows for a little bit more of an expansive understanding of connection… but I think kinship network is something that feels quite specific, and I think in some ways assumes, maybe not assumes, but to me there’s a little bit more of a culturally laden definition of connection in kinship”.—CI 6 “Is this the same question as above [item 2.5B]?… I think my preference is the second one”.—CI 8	Participants identified redundancy between this item and item 2.5B (*Through food I am part of a kinship network with plants, animals, land, water, and those around me*). As a result of participant preferences, item 2.5B was dropped and this item was retained.	Through food I feel connected to plants, animals, land, water, and others.
In the last year, how often did you participate in ceremony with other people in your urban Native community?	“I had access once or twice, but they were not necessarily ceremonies from my community, just sort of broadly indigenous the way that you do when you’re in urban area. Somebody has Lakota ceremony or something, and I don’t usually participate in that… Just like you have the opportunity to share your knowledge, you have the opportunity to go to ceremony, are you happy with that, or is it harming you?”—CI 7 “I know that it’s an important part of ceremony, but I think that for me personally, I think that due to Westernization and colonization, we as a whole have lost our connection to food and what actually is ancestral and what actually would have been traditionally, I guess, at ceremony because it wouldn’t be brisk iced tea and mountain dew and fry bread and stuff like that. So I think that food wise, unless it’s like a prayer plate, I don’t really feel that much more connected when it comes to ceremony and food”.—CI 8	This feedback combined with other feedback pointed to that access and satisfaction with access to Indigenous foodways can be two entirely different questions. Access questions were revised to reflect satisfaction with access to various Indigenous foodways. Ultimately, this item and the other item related to ceremony were dropped due to participant feedback that it was not significant for their typical connection to food.	Dropped (but access items in Measure C were reworded to reflect satisfaction with access)

## Data Availability

Data described in the manuscript, code book, and analytic code will not be made available because doing so would interfere with the data sovereignty of our urban AI/AN research partners.

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
