# Peer review of "Community-Engaged Development of Strengths-Based Nutrition Measures: The Indigenous Nourishment Scales"

_ijerph, 2024, doi:10.3390/ijerph21111496_

Round 1

Reviewer 1 Report

Comments and Suggestions for Authors

Dear Authors,

First of all, I would like to congratulate you on this important work. The manuscript and the overall purpose of the larger research are significant in the contributions they will make to not only the academy, but within Indigenous communities throughout Turtle Island.

I have just a few comments, questions and suggestions to consider, as follows:

In the Introduction, additional context would be helpful to frame the critiques associated with the more individualized form of food guidance that exist within the U.S. and Canada, along with the concept of nutritionism. For example, see:

Scrinis G. Nutritionism: The Science and Politics of Dietary Advice. Routledge; 2013.

Graham R, Stolte O, Hodgetts D, Chamberlain K. Nutritionism and the construction of ‘poor choices’ in families facing food insecurity. Journal of Health Psychology. 2017;23(14): 1863- 1871. doi: 10.1177/1359105316669879.

Ayo N. Understanding health promotion in a neoliberal climate and the making of health conscious citizens. Critical Public Health. 2012; 22(1): 99-105.

In the Materials and Methods section, it is not clear how the CRCs or Community Partners contributed towards the study design overall along with how many CRCs were created and what constituted their membership. Positionality of all research team members not included. Figure 1 is a helpful roadmap to the research process, but there are several instances where it is unclear who participated and how they were involved in the phases of data collection.

Please see additional comments, suggestions and questions in the attached version of the manuscript.

Author Response

Please see attached response to Reviewer 1

Reviewer 2 Report

Comments and Suggestions for Authors

The nourishment scales and their implementation will have far-reaching positive effect on better understanding of food use in AI/AN communities. Congratulations on innovative and very promising work!  This reviewer was initially skeptical about potential bias in the selection of the Community Research Councils and the other participating partners in the data collection.  It would be helpful to have a few more sentences to describe how the process of the purposive sampling excluded bias. However, Figure 2. and the results of Table1. demonstrating the results ring true for my understanding of AI/AN worldviews on food, diet and health.  

The need for further research is highlighted in lines 488-496. This is especially so for Indigenous Peoples in other national and international communities and their cultural and environmental settings.

please describe how the focus group members and the interviewees were selected. For example, what criteria were you searching for and how did you elicit this information from the research councils or the Indigenous leadership in the area?

I look forward to reading more reports using this research methodology in the peer-reviewed literature.

Author Response

Please see attached document for response to reviewer 2. 

Reviewer 3 Report

Comments and Suggestions for Authors

Reviewer Comments:

General: Thank you for the opportunity to review this strong paper. It is an excellent paper that brings a strengths-based approach into nutrition, where the focus is so often on deficits, thank you! I think this paper is ready for publication with some minor revisions. Primarily, I would strongly encourage the authors to consider some minor restructuring of the paper that I think would GREATLY help the reader! My other points below are simply observations that the authors can consider for this, or future work, as they find useful.

·       The only place where I would strongly urge the authors to consider revisions is around the structure of the paper, particularly the results. As the reader reads the paper, they are exposed to a lot of discussion about “the Indigenous Nourishment Model” and “the scales” but it is not until virtually the end of the paper that the reader finds out what these look like! My suggestion would be to present the final Model and the final scales as early as possible in the paper. The current structure was quite frustrating for me as a reader, I think I would have gotten more out of the very rich and interesting coverage of community feedback and the process in general, if I had been presented the Model and the Scales before the rest of the results. My suggestion would be to have a results section that goes: final Model; final Scales; process and community input. The community input and feedback are super interesting, and I really appreciated them (the process used is a significant strength of the paper) so I wouldn’t suggest removing them, so much as just changing the order the reader sees them.

·       Also, the Results include sections that I strongly recommend should be moved to the Methods section. Lines 253-258 = Methods instead of results. Lines 336-339 = Methods instead of results. Lines 390-395 = Methods

·       The authors talk about epistemology in a number of places, but I wonder if a comparison between Indigenous and western “ontology” (world view) might be more / also productive? For example, on line 113, the aspects of the INM you present include “relational” which is a central term/ concept in work on Indigenous Ontology (see for example Hunt 2014, but so many others). Noting that the “Ontological turn” in social science has also been critiqued by Indigenous scholars (Todd 2016).

o   Todd, Z. (2016). An indigenous feminist's take on the ontological turn: ‘Ontology’is just another word for colonialism. Journal of historical sociology29(1), 4-22.

o   Hunt, S. (2014). Ontologies of Indigeneity: The politics of embodying a concept. Cultural geographies21(1), 27-32.

·       A final reflection that I think is beyond revisions for this paper: I appreciated that the authors noted the role of this assessment tool relative to conventional tools. As someone who studies the cultural value of food and food culture as drivers of dietary choice (in the context of / in conjunction with food environments drivers), I do wonder if maybe what these nourishment scales are measuring is cultural attachment to foods/ attachment to cultural foods and food ways/ or maybe the ways food nurtures cultural wellbeing and identity, as opposed to “nourishment” in the conventional nutrition sense? I can see myself using some or all of these scales to assess attachment to cultural food practices as one aspect of food choice…. I am troubled with the conundrum of how to assess diet in a manner that doesn’t feel “western”, demeaning or othering to Indigenous communities, but at the same time has demonstrated associations with health outcomes that matter to us all? In a future work I would love to know if these scales are associated with health outcomes? Does that matter to the Native Communities you work with?

More Specific:

·       Line 64. You talk about Indigenous knowledge being disregarded, but for years it was denigrated, shamed, and governments actively worked to erase it, as part of the settler colonial project to sever links between communities and land. You could be more assertive about the harms done here.

·       Line 200. You use the acronym IDI which I am not familiar with even though in-depth interviews are a core part of my research practice. I had to go back and remind myself multiple times what IDI was! I suggest just using the full term "in-depth interviews".

·       Measure C, all the questions are “for the past year”, why 1 year?

·       Reference #1 appears to be the same as #3. I’m not sure about the journal reference format, but normally under a numbering system you should just re-use #1, as you have done in other cases in the paper?

Author Response

Please see the attached document for response to reviewer #3. 
